# The Conserved Cysteine-Rich Secretory Protein MaCFEM85 Interacts with MsWAK16 to Activate Plant Defenses

**DOI:** 10.3390/ijms24044037

**Published:** 2023-02-17

**Authors:** Ni Cai, Xiangqun Nong, Rong Liu, Mark Richard McNeill, Guangjun Wang, Zehua Zhang, Xiongbing Tu

**Affiliations:** 1The State Key Laboratory for Biology of Plant Diseases and Insect Pests, Institute of Plant Protection, Chinese Academy of Agricultural Sciences, Beijing 100193, China; 2AgResearch, Lincoln Science Centre, Private Bag 4749, Canterbury 8140, New Zealand

**Keywords:** wall-associated kinase, CFEMs, *Metahizium anisopliae*, plant immunity, *Medicago sativa*

## Abstract

*Metarhizium anisopliae* is an entomopathogenic fungus which may enhance plant growth and resistance when acting as an endophyte in host plants. However, little is known about the protein interactions nor their activating mechanisms. Common in fungal extracellular membrane (CFEM) proteins have been identified as plant immune regulators that suppress or activate plant resistance responses. Here, we identified a CFEM domain-containing protein, MaCFEM85, which was mainly localized in the plasma membrane. Yeast two-hybrid (Y2H), glutathione-S-transferase (GST) pull-down, and bimolecular fluorescence complementation assays demonstrated that MaCFEM85 interacted with the extracellular domain of a *Medicago sativa* (alfalfa) membrane protein, MsWAK16. Gene expression analyses showed that MaCFEM85 and MsWAK16 were significantly upregulated in *M. anisopliae* and *M. sativa*, respectively, from 12 to 60 h after co-inoculation. Additional yeast two-hybrid assays and amino acid site-specific mutation indicated that the CFEM domain and 52th cysteine specifically were required for the interaction of MaCFEM85 with MsWAK16. Defense function assays showed that JA was up-regulated, but *Botrytis cinerea* lesion size and *Myzus persicae* reproduction were suppressed by transient expression of MaCFEM85 and MsWAK16 in the model host plant *Nicotiana benthamiana*. Collectively, these results provide novel insights into the molecular mechanisms underlying interactions of *M. anisopliae* with host plants.

## 1. Introduction

Interactions between plants and microbes are widespread and are the result of past and ongoing co-evolution. These interactions can have either beneficial, neutral, or adverse outcomes for each participant [1,2,3]. Beneficial rhizospheric microbiota can enhance plant growth and improve overall health by protecting against soilborne diseases or enhancing nutrient uptake [4,5]. A wide range of beneficial microbes can enhance plant capabilities such as nutrient uptake, growth, and pathogen and insect defenses [6,7]. 

*Metarhizium* Sorokīn is an important genus of entomopathogenic fungi with the ability to colonize plants [8], but there is also evidence to suggest that members of the genus can enhance plant pathogen resistance. For example, a laboratory study found 60% inhibition of *Fusarium solani* (Mart.) Sacc. in the presence of *Metarhizium robertsii* (formerly known as *M. anisopliae*) compared with controls without *M. robertsii* [9]. Some *Metarhizium* strains have the potential to improve plant resistance to insect pests and diseases by altering the expression of plant defense genes. Endophytic colonization with *M. robertsii* activates expression of both the jasmonic acid (JA) and salicylic acid (SA) defense pathways in maize leaves; furthermore, such colonization is associated with the promotion of plant growth and suppression of *Agrotis ipsilon* (Hufnagel) larval development [10]. *Metarhizium guizhouense* activated the β-1,3-glucanase and chitinase in fruit peel of *Lansium parasiticum* resulting in the growth inhibition of *Botrytis* sp. and *Fusarium* sp. on the fruit of *Aglaia dookkoo* Griff [11]. Moreover, when *M. anisopliae* was applied to peanut seedlings, a variety of transcription factors including WRKYs, MYCs, TGAs, and ethylene-responsive transcription factors were activated, while nitrate transporters and proteins binding dehydration-responsive elements were also differentially expressed [12]. This suggests that *M. anisopliae* regulates plant defense genes as it colonizes the plant. However, relatively little is known about the mechanisms underlying plant defense responses following infection by *Metarhizium*.

Effectors are a common means by which microorganisms regulate plant resistance [13]. The activation of plant resistance is usually characterized by an oxidative burst and the induction of hormones, metabolites, and other signals [13]. The plant hormones SA and JA play important roles in inducing plant resistance, mediating two separate defense signaling pathways [14]. The SA signaling pathway primarily functions in the plant allergic response and in systemic acquired resistance to pathogens, but it may also be involved in indirect plant defense responses induced by stinging insect feeding [15]. Accumulation of SA is associated with increased expression of genes encoding lipid transfer proteins (LTPs) and phenylalanine ammonia-lyase (PAL) [16], which mediate the synthesis and accumulation of downstream specialized metabolites such as flavonoids and lignin [17]. The JA signaling pathway functions in direct and indirect responses to mechanical damage, fungal pathogens, and insect pests. Studies have shown that JA signaling has a regulatory effect on the synthesis of specialized metabolites such as terpenoids, phenylpropanoids, and alkaloids, which have a wide range of biological functions [18]. Some specialized metabolites have been shown to accumulate in plant cells following treatment with methyl-JA (MeJA), including paclitaxel in *Taxus* species [19,20], terpenoids in *Centella asiatica* (L.) Urban [21], and saponins in ginseng [22]. Application of SA and chitosan (CHT) as elicitors induced lignin accumulation and defense enzyme reactions in tomato, which reduced the incidence of *Ralstonia solanacearum* (Smith) Yabuuchi infection [23]. In addition, JA and SA levels in wheat were significantly accumulated by application of the elicitor PeaT, enhancing the defense response to the oat aphid *Sitobion avenae* (Fabricius) [24]. This indicates that plant immunity could be regulated by these effectors through plant hormones and metabolites.

Common in fungal extracellular membrane (CFEM) proteins are present in a wide range of fungi. They encode domains that are usually 60 amino acids (aa) in length and contain eight characteristically spaced cysteines [25]. CFEM domains are similar to the epidermal growth factor (EGF)-like domains, which function as extracellular receptors, signal transducers, or adhesion molecules in host–pathogen interactions [26]. Proteins containing CFEM domains may manipulate the plant resistance response by acting as effectors. In the wheat leaf rust fungus *Puccinia triticina* Erikas, the CFEM effector candidate PTTG_08198 accelerated the progress of cell death and promoted reactive oxygen species (ROS) accumulation [27]. The anthracnose fungus *Colletotrichum graminicola* (Ces.) G. W. Wils contains five effectors (CgCFEM6, 7, 8, 9, and 15) that have been shown to suppress BAX-induced programmed cell death in *Nicotiana benthamiana* Domin [28]. The phytosymbiotic mycorrhizal fungus *Laccaria bicolor* (Maire) P.D.Orton was reported to secrete several CFEM proteins, such as Lac310796, Lac296573, and Lac296572, in symbiotic tissues [29]. Although the complex functions of these proteins have not been studied further in mycorrhizal symbiosis, prior results suggest that CFEM proteins may function in signaling between fungi and plants.

*M. anisopliae* can function as either an entomopathogenic or endophytic fungus in a host plant [30]. However, the roles of CFEM proteins have not yet been reported. We previously identified a CFEM domain-containing protein in *M. anisopliae*, MaCFEM85 (GenBank ID: MZ682609) [31]. Here, we report on the relationship between MaCFEM85 and the *Medicago sativa* wall-associated kinase MsWAK16. We analyzed the MaCFEM85 sequence and conducted experiments to determine which residues are critical for the interaction with MsWAK16. Finally, using *N. benthamiana* as our model plant, we evaluated plant defense responses to *Botrytis cinerea* and the aphid *Myzus persicae* with and without transient expression of MaCFEM85 and MsWAK16.

## 2. Results

### 2.1. MaCFEM85 Was Most Closely Related to Non-Pathogenic Fungal CFEMs and Localized to the Plasma Membrane

A phylogenetic tree was built to analyze the relationships between MaCFEM85 and other CFEM proteins from pathogenic and non-pathogenic model fungi. These included *Magnaporthe oryzae*, *Fusarium oxysporum* (Schl)., *Neurospora crassa* Shear & B.O.Dodge, and *Beauveria bassiana* (Bals.) (see Section 4). MaCFEM85 was most closely related to CFEM proteins in *B. bassiana*, and therefore, evolutionarily closer to non-pathogenic than to pathogenic fungi (Figure 1a).

We conducted heterologous expression analysis in *N. benthamiana* to investigate the subcellular localization of MaCFEM85 at the cellular level. When the test leaves were infiltrated with MaCFEM85–eGFP and MaCFEM85-mcherry fusion protein, orange fluorescence was detected in the plasma membrane, that is contrast to eGFP and mCherry co-expression, which orange fluorescence was detected in both the plasma membrane and nuclei of leaves (Figure 1b). This suggests that MaCFEM85 plays a role in the cell membrane.

### 2.2. MaCFEM85 Was Upregulated during Interactions with M. sativa

Wall-associated receptor-like kinases (WAKs) represent a subgroup within the receptor-like protein kinases (RLKs) superfamily, and include an extracellular domain, a transmembrane helix, and an intracellular kinase domain. They play an important role in regulating plant growth, development, stress response and pathogen resistance signaling pathways [32]. RNA was extracted from *M. anisopliae* hyphae and *M. sativa* roots after co-incubation to investigate the expression patterns of MaCFEM85 and MsWAK16. Both MaCFEM85 and MsWAK16 were expressed at significantly higher levels during co-incubation from 12 to 60 h (Figure 1c,d). From 0 to 36 h, MaCFEM85 expression gradually increased, peaking at 36 h. It was × 593 times more highly expressed in the combination of *M. anisopliae* and *M. sativa* compared with *M. anisopliae* grown alone. Although MaCFEM85 expression was slightly decreased in the 48–60 h period, it was still expressed at significantly higher levels than in *M. anisopliae* grown alone. MsWAK16 was also significantly upregulated, with expression peaking at 36 h. In *M. sativa* infected with *M. anisopliae*, MsWAK16 was × 89.06 times more highly expressed than in *M. sativa* without *M. anisopliae*. From 36 to 60 h, levels of MsWAK16 decreased slightly, but its expression was still significantly higher than in the uninoculated plants. 

### 2.3. MaCFEM85 Interacts with MsWAK16 In Vitro and In Vivo

To investigate the potential functional mechanism of MaCFEM85 in response to *M. anisopliae* treatment, a yeast two-hybrid (Y2H) screen was performed to preliminarily identify host proteins that interact with MaCFEM85. The interaction between the extracellular domain of MsWAK16 (MsWAK16-ED) and MaCFEM85 was examined with a one-to-one yeast two-hybrid assay using MaCFEM85 in pGBKT7 as the BD vector and MsWAK16 in pGADT7 as the AD vector. All of the transformed yeast grew well on SD-T/L deficient medium, and the positive control group and experimental group grew successfully on SD-T/L/H/A + X-α-gal deficient medium. This indicated that MaCFEM85 could interact with MsWAK16-ED (Figure 2a). 

For in vitro validation, glutathione-S-transferase (GST)-tagged *MsWAK16*-ED (encoding a product of 60 kDa) was inserted into pGEX6-P2, and polyhistidine (His)-tagged *MaCFEM85* (encoding a product of 17 kDa) was inserted into pET-21b. Immunoblotting with His and GST antibodies showed that the recombinant protein MaCFEM85-His interacted with the GST-MsWAK16-ED prey but not with GST alone, and that GST-MsWAK16-ED interacted with His-MaCFEM85 (Figure 2b).

To further confirm the interaction between MaCFEM85 and MsWAK16, we performed a bimolecular fluorescence complementation (BiFC) assay with MaCFEM85-YFP^N^ and MsWAK16-YFP^C^ constructs. Co-expression of MaCFEM85-YFP^N^ and MsWAK16-YFP^C^ in tobacco leaves generated a yellow fluorescence signal, indicating that MaCFEM85 interacted with MsWAK16 (Figure 2c).

### 2.4. Key Sites for MaCFEM85 and MsWAK16 Interactions

To define the region of MaCFEM85 that is required for the interaction with MsWAK16-ED, protein domains were predicted with the online software SMART (Figure 3a). MaCFEM85 contained a CFEM domain the 19-86 aa region. The tertiary structure was also predicted (Figure 3b). The eight cysteines in this domain resulted in four disulfide bonds (CYS26 and CYS69, CYS30 and CYS64, CYS43 and CYS50, and CYS52 and CYS85), maintaining stability of the protein (Figure 3b). To validate the putative interaction site(s) in MaCFEM85, seven mutated versions of the protein were generated and inserted into pGBKT7 as bait proteins. Each recombinant vector was co-transfected with AD-MsWAK16-ED in the Y2H Gold strain. The strain with CYS52 mutated did not grow on selective media (Figure 3c, outlined in red), indicating that CYS52 is required for the interaction of MaCFEM85 with MsWAK16-ED.

### 2.5. The Interaction of MaCFEM85 with MsWAK16 Activates the Plant Immune Response

#### 2.5.1. Evaluating the Role of MaCFEM85 and MsWAK16 in Disease Resistance against *B. cinerea*

To explore the possible involvement of MaCFEM85 and MsWAK16 in pathogen defense responses, we examined whether overexpression of MaCFEM85, MsWAK16, or MaCFEM85 and MsWAK16 could confer increased resistance to *B. cinerea*. We transiently expressed these vectors in *N. benthamiana* leaves. A Western blot assay showed that MaCFEM85 and MsWAK16 were expressed at comparable levels when they were expressed alone or together, which shows that MaCFEM85 and MsWAK16 were successfully expressed in tobacco (Figure 4a). Disease assays were also performed using *N. benthamiana* infiltrated with MaCFEM85, MsWAK16, MaCFEM85+MsWAK16, or the eGFP control. Lesions were significantly smaller (by ~30% at 2 d after inoculation) on leaves infiltrated with MaCFEM85, MsWAK16, or MaCFEM85+MsWAK16 compared with the control plants (Figure 4b). These data demonstrate that transient expression of MaCFEM85 in *N. benthamiana* conferred increased resistance to *B. cinerea* and that MaCFEM85 and MsWAK16 positively regulated the defense response against *B. cinerea*.

#### 2.5.2. Evaluating the Role of MaCFEM85 and MsWAK16 in Aphid Defense in *N. benthamiana*

To further investigate the role of MaCFEM85 and MsWAK16, *N. benthamiana* plants transiently expressing MaCFEM85 and MsWAK16 were infested with *M. persicae* and the populations were assessed. For this experiment, a large area of each *N. benthamiana* leaf was agroinfiltrated with the recombinant binary vector pYBA1132 containing *MaCFEM85* and *MsWAK16*. GFP was used as the control. At 12 h after infiltration, 20 *M. persicae* adults were caged on each leaf, exposing the infiltrated area to aphids. On the following three days, adult aphid mortality and the number of nymphs were recorded; the nymphs were then removed. There was no significant difference in the mortality risk amongst *M. persicae* populations feeding on plants expressing MaCFEM85, MsWAK16, or MaCFEM85+MsWAK16 (χ2 = 3.65827, DF = 3, *p* < 0.30081) (Figure 4c). However, at 24, 48, and 72 h, the average number of progeny produced by each adult aphid was significantly higher on *N. benthamiana* expressing the GFP control compared with those expressing MaCFEM85, MsWAK16, or MaCFEM85+MsWAK16 (Figure 4d).

#### 2.5.3. Evaluating the Role of MaCFEM85 and MsWAK16 in Hormone Accumulation and Hormone-Related Gene Expression

To analyze the differences between the defense responses of *N. benthamiana* plants expressing eGFP, MaCFEM85, MsWAK16, and MaCFEM85+MsWAK16, we measured JA, SA, and total flavonoid levels, and the expression of related genes. The results showed that JA and SA levels differed between plants expressing eGFP, MaCFEM85, MsWAK16, and MaCFEM85+MsWAK16 (Figure 5a). Compared with those expressing eGFP, JA levels were significantly lower in plants expressing MaCFEM85; SA levels were slightly increased, but the differences were not significant. In contrast, plants expressing MsWAK16 showed no significant differences in JA levels compared with the eGFP control, whereas SA levels were significantly lower than in the control (Figure 5a). JA and SA levels were significantly increased and decreased, respectively, in plants expressing MaCFEM85+MsWAK16 compared with eGFP. 

Total flavonoid content was significantly higher in plants expressing MaCFEM85+MsWAK16 compared with all other treatment groups (Figure 5a). The biosynthetic gene expression levels were similar in plants expressing MsWAK16 and MaCFEM85+MsWAK16. For example, genes related to the JA response, namely *COI1*, *MYC2,* and *PDF1.2*, were significantly upregulated compared with plants expressing eGFP (Figure 5b). Similarly, the SA-related genes *NPR1*, *WRKY70*, and *PR1* were significantly differentially expressed in plants expressing MsWAK16 and MaCFEM85+MsWAK16; *NPR1* was upregulated, whereas *WRKY70* and *PR1* were downregulated compared with the control (Figure 5b). 

We also examined the expression of key genes in the Shikimic acid and phenylpropanoid synthesis pathways, which are both related to flavonoid synthesis. In general, expression of MaCFEM85 alone did not induce significantly higher expression of these genes in tobacco. However, these genes were upregulated in tobacco expressing MsWAK16 or MsWAK16+MaCFEM85 compared with the eGFP control (Figure 5c).

## 3. Discussion

### 3.1. The Conserved CFEM Protein Motif Serves Multiple Functions in Fungal Species 

The CFEM domain is unique to fungi and commonly occurs in fungal extracellular membrane proteins. The domain originated in the most recent common ancestor of Ascomycota and Basidiomycota [33]. Recent research has shown that CFEM proteins in fungal pathogens can act as plant immune regulators, causing host plant immune suppression or activation depending on the type of infection [28,34,35]. There have been few reports of CFEM domain-containing proteins in *M. anisopliae*, only in the context of an evolutionary comparison with other fungi [36]. In this study, we compared *M. anisopliae* CFEM proteins with other known CFEM proteins in both pathogenic and non-pathogenic fungi. We confirmed that the closest homolog of MaCFEM85 is Cfem5 found in *Beauveria bassiana*, one of 12 CFEM proteins in that species (Appendix A). BbCfem5 is essential for iron acquisition [37]. Moreover, based on the predicted tertiary structure, the CFEM domain in MaCFEM85 is very similar to Surface Antigen Protein 2 (CSA2) found in *Candida albicans* (C.P.Robin) Berkhout, in which 65 residues (96% of the sequence) have been modelled with 99.8% confidence [31]. *Candida albicans* is an animal pathogen; CSA2 plays an important role in growth and pathogenicity by extracting heme from hemoglobin and transporting heme from the cell wall to the plasma [38,39,40]. We hypothesis that MaCFEM85 may be involved in *M. anisopliae* virulence of insect hosts. These combined functions of MaCFEM85 in animal pathogenic infection and in activation of plant immunity make the protein an exciting future research focus. 

### 3.2. Disulfide Bonds Are Important Structures for Protein Function

The conformational integrity of a protein structure is directly related to its ability to function. The disulfide bonds formed by cysteine pairs promote protein folding and conformational stability, and are considered to form key sites for recognition and binding of specific receptors or ligands [41]. For example, disrupting any one of the three conserved disulfide bonds in the plant pathogen *Cladosporium fulvum* effector protein AVR4 results in protease sensitivity and reduced chitin binding ability [42]. In three other *C. fulvum* proteins (ECP1, ECP2, and ECP5), single substitutions of alanines for cysteines dampen the tomato hypersensitive response, indicating that the cysteines are critical to maintaining stability and hypersensitive response-inducing activity [43]. Moreover, in the mature protein MC69 of *Magnaporthe oryzae*, mutagenesis of two conserved cysteine residues (Cys36 and Cys46) may impair MC69 function without affecting secretion, suggesting the importance of the disulfide bond specifically in the pathogenicity of MC69 [44]. The CFEM domain appears to be similar in size and in the pattern of cysteine residues to EGF-like domains, which contain three or four pairs of disulfide bonds. EGF proteins can function as cell-surface receptors, signal transducers, or adhesion molecules in host–pathogen interactions [45]. In this study, we not only found that the CFEM domain of MaCFEM85 contained eight conserved cysteines that formed four pairs of disulfide bonds (Figure 3b), but we also validated that CFEM was the critical domain for the interaction between MaCFEM85 and MsWAK16. Furthermore, through site-directed mutation of cysteine residues to alanine, we found that the cysteine residue at position 52 was the core site required for the interaction (Figure 3c). These results provide a basis for further studies of the physiological functions of the CFEM85–WAK16 interaction.

### 3.3. The Interaction of MaCFEM85 with MsWAK16 Activated Plant Defenses

In microbial–plant interactions, plant defense responses are generally activated by microbial effector proteins. Induced defense is becoming an important tool in biological pest control to promote resistance. CFEM proteins have been identified as effectors involved in the regulation of plant immune activation or inhibition [28,46]. However, there is a paucity of published research linking the regulation of these immune factors to their specific roles in plant disease and insect resistance. In this study, we used an entomopathogenic and endophytic fungus, *M. anisopliae*, to identify an interaction between MaCFEM85 and a cell wall-associated kinase, MsWAK16, in *M. sativa.* The results showed that this interaction reduced the lesion ratio after inoculation with *B. cinerea* (Figure 4b) and decreased the population growth rate of *M. persicae* (Figure 4d), indicating the interaction between MaCFEM85 and MsWAK16 increased *M. sativa* resistance to aphids.

Changes in the levels of JA, SA, and ethylene (ET) can be used as markers to evaluate the induction of plant resistance [47,48]. Here, we demonstrated that MaCFEM85 interacted with MsWAK16 to influence plant resistance through hormonal regulation and to inhibit the reproductive rate of *M. persicae* (Figure 4d). Similar studies have been reported previously. For example, the *Brevibacillus laterosporus* effector protein PeBL1 was shown to induce JA and SA accumulation in tomato and to decrease the growth rate of the second and third generation populations of *M. persicae* that fed on those plants. Moreover, tomato plants sprayed with effectors have a repellent effect on *M. persicae* [49]. Application of the elicitor protein PeBC1 to common bean (*Phaseolus vulgaris* L.) leads to pronounced and significant sub-lethal effects on green peach aphids. Plants treated with PeBC1 show significant upregulation of genes related to JA and SA [50]. The *Beauveria bassiana* elicitor PeBb1 reduces the fecundity of *M. persicae* on tobacco and induces the expression of JA- and ET-related genes [51]. In the present study, transient expression of MaCFEM85 and MsWAK16 in tobacco significantly upregulated the JA response-related genes *COI1*, *MYC2*, and *PDF1.2* (Figure 5b) and increased JA and total flavonoid content (Figure 5a), which was comparable to results in the literature as described above. However, we did not detect high levels of SA in tobacco 12 h after infiltration, and plants transiently expressing MsWAK16 or MaCFEM85+MsWAK16 showed significantly reduced SA levels and downregulation of genes involved in the SA response (Figure 5a,b). This demonstrated that stimulation of different plant hormones can lead not only to synergistic activities, but also to crosstalk and feedback, allowing plants to respond appropriately to different stimuli. Increased JA levels but low SA levels have been previously reported in plants. For instance, in *A. thaliana* defective for SA accumulation, JA levels were 25-fold higher than in wild type *A. thaliana*, and JA-responsive genes were activated [52]. It has also been reported that some bacteria can increase JA levels in plants to inhibit SA accumulation, avoiding the harm caused by SA. For example, *Pseudomonas syringae* targets the COI1 receptor through a toxin, coronatine, that negatively regulates the JA pathway [53]. This promotes MYC2-induced upregulation of the transcription factors ANAC019, ANAC055, and ANAC072 [54], inhibiting transcription of ICS1 (a key gene for SA synthesis) and thus downregulating SA production and signal transduction. In the present study, the interaction between MaCFEM85 and MsWAK16 led to increased JA accumulation and upregulation of JA-responsive genes but inhibition of SA accumulation and transcription of SA-related genes. This indicated that the interaction between MaCFEM85 and MsWAK16 induced JA, thereby improving plant resistance to aphids.

In *N. benthamiana* transiently expressing MaCFEM85 and MsWAK16, JA levels were increased and *B. cinerea* lesion sizes were decreased (Figure 4b), consistent with previous studies. JA is involved in plant resistance to insects and necrotrophic pathogens [52,55]. As a necrotrophic pathogen, *B. cinerea* was inhibited by JA and ET accumulation [56]. In the *A. thaliana* ET-deficient mutant *ein2-1* and the JA-response mutant *coi1-1*, levels of the downstream defense gene *PDF1.2* were significantly reduced and sensitivity to *B. cinerea* was enhanced [57]. We found here that JA accumulation and transcription of JA-related genes were significantly increased in *N. benthamiana* transiently expressing MaCFEM85 and MsWAK16, whereas the accumulation of SA and transcription of SA-related genes were inhibited. MaCFEM85 and MsWAK16 expression also showed an inhibitory effect on *B. cinerea*. This indicated that JA was activated by MaCFEM85 and MsWAK16 and played a leading role in plant resistance to *B. cinerea*.

## 4. Materials and Methods

### 4.1. Fungal Strains, Plant Materials, and Culturing Methods 

*Metarhizium anisopliae* isolate strains Ma 9 were cultured on potato-sugar-agar (PSA) medium (200 g peeled potato extract boiled in water, 20 g sucrose, and 15 g agar/L). Fresh sporangium powder was collected after 10 d. *Nicotiana benthamiana* plants were grown in an artificial climate chamber with a 14/10 h light/dark cycle (27/25 °C). For transient gene expression via Agrobacterium-mediated transformation, *Agrobacterium tumefaciens* strain GV3101 was cultured in Luria Broth (LB) medium (10 g tryptone, 5 g yeast extract, and 10 g NaCl/L). Yeast strain Gold (OE biotech Co., Ltd., Shanghai, China) was cultured on yeast extract peptone dextrose (YPDA) medium (10 g yeast extract, 20 g peptone, 20 g glucose, and 0.03 g adenine hemisulfate/L). For each vector and strain, appropriate antibiotics were used, namely rifampin, kanamycin, or ampicillin (25, 50, or 50 µg/mL, respectively). The strains and plasmids used in this study are listed in Appendix A.

*Medicago sativa* seeds were surface sterilized with 75% ethanol for 1 min, followed by 50% NaClO (5.5%) for 15 min. The seeds were thoroughly mixed, then washed in sterile water three times for 5 min each. The seeds were incubated at 4 °C in the dark for over 24 h, then germinated on 1% water agar plates at room temperature overnight. Three days after gemination, the developing seedlings were transferred to filter paper on 9-cm sterile petri dishes, with 20 seedlings per dish. The treatments and control were allocated to 15 dishes each, respectively. The seedlings were then irrigated with 10 mL of *M. anisopliae* spore suspension containing 10^7^/mL, and the control was irrigated with 10 mL sterile water. At 0, 12, 24, 48, and 60 h, a random selection of seedlings was taken from three petri dishes for each time interval. The samples were frozen in liquid nitrogen and stored at −80 °C.

### 4.2. qRT-PCR and Plasmid Construction

Total RNA was extracted from *M. anisopliae* (hyphae) and *M. sativa* (roots) with TRIzol reagent (Invitrogen, CA, USA) following the manufacturer’s instructions. The quality and abundance of the resulting RNA were measured with a NanoPhotometer^®^ (Implen, Münich, Germany). First-strand cDNA synthesis (up to 2ug RNA) was performed using 5× All-In-One RT Master Mix (Applied Biological Materials Inc., Vancouver, BC, Canada) following the manufacturer’s instructions. 

qRT-PCR was performed using 2×SYBR Green qPCR Master Mix from US EVERBRIGHT ^®^ INC. and the ABI QuantStudio 5 system following the manufacturer’s instructions. The primers used for each gene are listed in Appendix A. *MaTry* [58], *Msactin* [59], and *Nbactin* [60] were used as the internal reference genes for normalization of expression data. The reaction was performed under the following conditions: 5 min at 95 °C, followed by 40 cycles at 95 °C for 15 s and at 60 °C for 40 s. There were three technical replicates for each sample. Relative expression was calculated using the 2^−ΔΔCt^ method [61]. Statistical significance was determined using one-way analysis of variance (ANOVA) and Duncan’s multiple-range test in SPSS v20.0 (SPSS).

### 4.3. Transient Expression of Proteins in N. benthamiana 

Ma*CFEM*85 coding sequence (CDS) was amplified with ultra-fidelity DNA polymerase using cDNA as the template.

For the subcellular localization assays, PCR products containing the CDS for MaCFEM85 were cloned into pYBA1132-eGFP (digested with *EcoR*I and *Sal*I) and pcambia1300-mcherry (*EcoR*I and *Sac*I), respectively. All the constructs were validated with sequencing (Tsingke Biotechnology Co., Ltd. Beijing, China). The primers used in this study are listed in Appendix A.

For transient expression of MaCFEM85-eGFP and MaCFEM85-mCherry in *N. benthamiana*, *A. tumefaciens* strain GV3101 was transformed with each plasmid and verified with PCR. The Agrobacterium was cultured overnight at 28 °C with shaking. The cells were collected via 5 min 5000 rpm centrifugation at room temperature, washed three times with sterile double-distilled water, and resuspended in 10 mM MgCl_2_ buffer (containing 10 mM MES and 10 mM acetosyringone, pH 5.7). The cell suspension was adjusted to an OD_600_ of 0.5, then infiltrated into the underside of 4- to 5-week-old *N. benthamiana* leaves with a 1-mL syringe. Each leaf was infiltrated with 50 µL *A. tumefaciens*; three leaves were treated per plant and three plants were in each treatment group. The treated leaves were collected after 30 h and visualized with a Zeiss LSM980 confocal microscope (Carl Zeiss, Germany) to determine the sub-cellular localization.

### 4.4. Phylogenetic Analysis

A phylogenetic tree was constructed using the neighbor-joining method in MEGA X. There were 1000 bootstrap replicates using the P-distance model. The amino acid sequences used to generate the tree were obtained with BLASTP searches against the NCBI database of related fungi (e.g., *Magnaporthe oryzae*, *Botrytis cinerea*, *Fusarium graminearum*, *Colletotrichum graminicola*, *Fusarium oxysporum*, *Neurospora crassa*, *Aspergillus fumigatus*, *Lasiodiplodia theobromae*, *Gliocladium roseum*, *Trichoderma harzianum*, *Beauveria bassiana*, and *Paecilomyces lilacinus*) using MaCFEM85 as the query.

### 4.5. Yeast Two-Hybrid Assays

The Matchmaker Gold Yeast Two-Hybrid System (Clontech Laboratories, Inc.; now Takara Bio USA, Inc.) was used to verify the interaction between MaCFEM85 and MsWAK16. MaCFEM85 without the signal peptide was introduced into pGBKT7 as the bait and the extracellular domain of MsWAK16 (MsWAK16-ED) was inserted into pGADT7 as the prey. Yeast competent cell preparation and transformations were performed using a Frozen-EZ Yeast Transformation II Kit™ (ZYMO Research, CA, USA) following the manufacturer’s instructions. The bait and prey plasmids were co-transformed into the yeast strain Gold. Protein–protein interactions were analyzed based on growth on SD double dropout medium (DDO, SD/-Trp-Leu) and SD quadruple dropout medium (QDO, SD/-Trp-Leu-His-Ade) plates.

### 4.6. BiFC Assay

To generate the BiFC constructs, the pUC-SPYNE and pUC-SPYCE vectors were linearized by digestion with BamHI. The full-length CDSs of MaCFEM85 and MsWAK16 were each cloned and inserted into the linearized plasmids using a recombinant enzyme to obtain the MaCFEM85-YFP^N^ and MsWAK16-YFP^C^ constructs. The plasmids containing MaCFEM85 and MsWAK16 were co-transformed with empty vectors as negative controls (MaCFEM85-YFP^N^ + pUC-SPYCE, MsWAK16-YFP^C^ + pUC-SPYNE, and pUC-SPYNE + pUC-SPYCE). All the vectors were introduced into *N. benthamiana* via Agrobacterium-mediated transformation as described above. Fluorescence signals were observed in leaf epidermal cells using a Zeiss LSM980 confocal microscope (Carl Zeiss, Germany). The primers used for vector construction are listed in Appendix A.

### 4.7. GST Pull-Down Assay

For the GST pull-down assays, MsWAK16-ED was inserted into the pGEX-6P-2 vector and MaCFEM85 was inserted into pET-21b. The purified fusion proteins (GST- MsWAK16-ED) and the pGEX-6P-2 no-load protein (GST) were used as the bait protein and purified pET-21b-MaCFEM85 fusion protein (His-MaCFEM85) was used as the prey. GST pull-down assays were performed with the Mag-Beads GST Fusion protein purification system (Sangon Biotech, Shanghai, China) following the manufacturer’s instructions. Briefly, Mag-Beads were washed five times with 1× phosphate-buffered saline (PBS, pH 7.4) to release the alcohol protector, and 10 mL of GST or GST-MsWAK16-ED was then added. The beads were mixed by inversion at room temperature for 30 min. After removing the supernatant, the Mag-Beads were washed five times with 1× PBS. His-MaCFEM85 was added to Mag-Beads already bound with GST, and the mixture was incubated overnight at 4 °C with rotation. The beads were then washed five times with 1× PBS to remove unbound proteins. Subsequently, proteins immobilized on the beads were separated by SDS-PAGE, transferred to a nitrocellulose membrane (100 V, 1 h) and analyzed via Western blot. The membranes were washed three times for 10 min each with PBS + Tween (PBST). The membranes were blocked for 2 h at room temperature with 5% skimmed milk, then incubated with ProteinFind anti-His mouse monoclonal antibody (TransGen Biotech, Beijing, China) (diluted 1:3000) or ProteinFind anti-GST mouse monoclonal antibody for 2 h at 4 °C. The membranes were then immersed in ProteinFind goat anti-rabbit IgG(H+L) (HRP) antibody (TransGen Biotech, Beijing, China) (diluted 1:5000) for 1 h at room temperature. The membranes were visualized with the EasySee^®^ Western Blot Kit (TransGen Biotech, Beijing, China) following the manufacturer’s instructions.

### 4.8. Identification of the Key Site in MaCFEM85 

To determine the site required for the interaction of MaCFEM85 with MsWAK16, PCR amplification was performed to generate multiple truncated forms of MaCFEM85. The CFEM domain (MaCFEM85-CFEM; aa residues 19–86) and the C terminal without the CFEM domain (MaCFEM85-C, aa residues 87–170) were inserted into the vector pGBKT7 as the bait protein (Appendix A). Another five variants were constructed using polypeptide synthesis to mutate cysteine to alanine at positions 26, 30, 43, 52, and 26/30/43/50/52/64/69/85 (ΔCFEM85_26_, ΔCFEM85_30_, ΔCFEM85_43_, ΔCFEM85_52_, and ΔCFEM85_8 all_, respectively). These variants were used to perform Y2H experiments with MsWAK16-ED. The transformed yeast cells were assayed for growth on synthetic dropout SD/-Trp-Leu plates and SD/-Trp-Leu-His-Ade plates containing X-α-Galactosidase (X-α-Gal).

### 4.9. Plant Resistance Assays

*Myzus persicae* were obtained from Henan Quanying Biological Co., Ltd. One week before the start of bioassays, 150 adult aphids were placed on three *N. benthamiana* plants (50 aphids per plant). After 72 h, all the adults were removed with a soft artist’s brush, and the nymphs were allowed to feed for an additional four days before being transferred to *N. benthamiana* for the aphid performance assays. 

### 4.10. Aphid Performance Assays

*Nicotiana benthamiana* Domin. (Solanales: Solanaceae) was used to assess *M. persicae* performance, plant disease resistance against *B. cinerea*, and expression of hormone-related genes. Agrobacterium carrying either pYBA-eGFP, pYBA-MaCFEM85, pYBA-MsWAK16, or pYBA-MaCFEM85+pYBA-MsWAK16 were grown in LB supplemented with appropriate antibiotics for 36 h at 28 °C. The cells were washed three times, then resuspended in infiltration buffer (10 mM MgCl_2_, 10 mM MES, and 100 µM acetosyringone, pH 5.6) to an OD_600_ of 0.6. For co-infection, bacteria carrying MaCFEM85 and MsWAK16 were adjusted to an OD_600_ of 1.2, then mixed in equal volumes. Fully expanded leaves were infiltrated with bacteria using 1-mL needleless syringes. Three leaves were infiltrated on each plant and each treatment was applied to three plants for a total of 12 plants per experiment. 

For the aphid performance assays, 20 adult aphids were applied to the infiltrated area of each leaf 12 h after the Agrobacterium was applied. The aphids were confined using 5 mm diameter clip cages. Each treatment was repeated five times. Adult aphid mortality and the number of newly laid nymphs were recorded daily for three days.

*B. cinerea* was cultured for five days on PDAY. At 12 h after Agrobacterium infiltration, the newly cultivated *B. cinerea* was punched into a 5-mm fungus cake, and the mycelium growth surface was attached to the infiltration area on the leaf surface. To facilitate disease development, the plants were kept humid by covering with plastic film in trays at 22 °C to facilitate disease progression. After 48 h, disease progress was estimated in inoculated leaves by measuring lesion sizes. 

For quantifying the relative expression of relevant genes, the three infiltrated leaves were collected from each plant 12 h after infiltration, then frozen in liquid nitrogen and stored at −80 ℃. Total RNA extraction, cDNA synthesis, and qRT-PCR were conducted as described in Section 3.2. 

Salicylic acid (SA), jasmonic acid (JA), and flavonoid levels were measured in 15 infiltrated *N. benthamiana* leaves per treatment. The leaves were collected, frozen in liquid nitrogen, then stored at −80 ℃. Plant hormones were quantified using the Plant Salicylic Acid and Plant Jasmonic Acid ELISA Kits (Beijing WeLab Scientific Co., Ltd.), and flavonoids were quantified using the Micro Plant Flavonoids Assay Kit (Beijing Solarbio Science & Technology Co., Ltd.) following the manufacturer’s instructions. 

### 4.11. Statistical Analysis

Data were analyzed in SPSS v20.0. Significant differences in expression levels of MaCFEM85 and MsWAK16 were determined using Student’s *t*-test. Significant differences in SA, JA, and flavonoid levels were determined using one-way ANOVA and Duncan’s multiple-range test with a threshold of *p* < 0.05.

## 5. Conclusions

In this study, we identified and characterized a novel secreted protein, MaCFEM85, in *M. anisopliae*. It was found to be a conserved effector that can interact with MsWAK16 and activate the *N. benthamiana* defense response. We showed that the CFEM domain and the cysteine residue at position 52 in MaCFEM85 were critical for the interaction. This interaction may activate JA-related immune responses and disease-resistant and insect-resistant mechanisms in the plant.

## Figures and Tables

**Figure 1 ijms-24-04037-f001:**
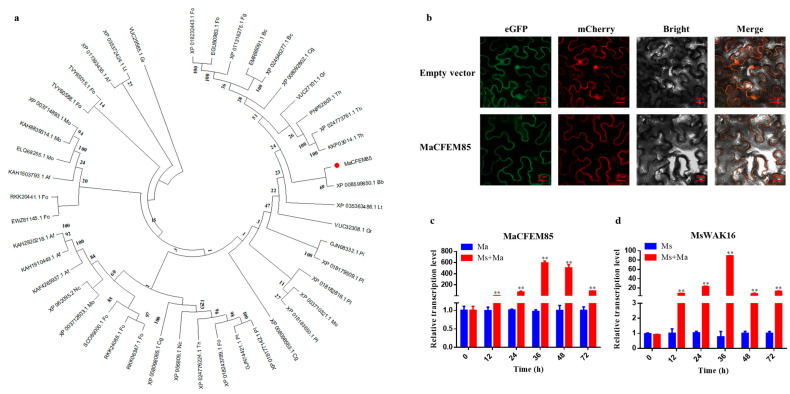
Phylogenetic analysis, subcellular localization, and expression patterns of MaCFEM85 and MsWAK16. (**a**) Neighbor-joining phylogenetic tree showing the relationships between MaCFEM85 and CFEM proteins in other fungi. The phylogenic tree was constructed using MEGA X. Mo, *Magnaporthe oryzae*; Bc, *Botrytis cinerea*; Fg, *Fusarium graminearum*; Cg, *Colletotrichum graminicola*; Fo, *Fusarium oxysporum*; Nc, *Neurospora crassa*; Af, *Aspergillus fumigatus*; Lt, *Lasiodiplodia theobromae*; Gr, *Gliocladium roseum*; Th, *Trichoderma harzianum*; Bb, *Beauveria bassiana*; Pl, *Paecilomyces lilacinus*. (**b**) Subcellular localization of MaCFEM85 proteins. The MaCFEM85–GFP fusion genes were co-expressed with the MaCFEM85-mCherry fusion genes in *N. benthamiana* leaves. The control vector carried eGFP and mcherry driven by the 35S promotor. Photos were taken at 30 h after infiltration with laser scanning confocal microscopy of eGFP (488 nm excitation and 507 nm emission), mcherry (587 nm excitation and 610 nm emission), bright field microscopy, and merged confocal and bright field. (**c**,**d**) qRT-PCR analysis of MaCFEM85 and MsWAK16 expression in co-incubated *M. anisopliae* and *M. sativa*. Ma, *M. anisopliae*; Ms, *M. sativa*; Ma+Ms, co-incubated Ma and Ms. Error bars show the standard deviation of three biological replicates. ** *p* < 0.01 (one-way ANOVA with Duncan’s multiple-range *t*-test).

**Figure 2 ijms-24-04037-f002:**
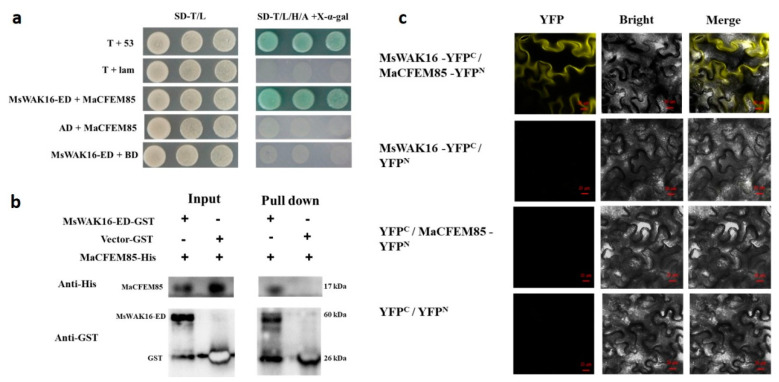
Validation of the interaction between MaCFEM85 and MsWAK16. (**a**) Verification of the interaction between AD-MsWAK16-ED and BD-MaCFEM85 in Y2H Gold yeast. pGBKT7-53 and pGADT7-RecT (T+53) and pGBKT7-lam and pGADT7-RecT (T+Lam) were used as the positive and negative controls, respectively. (**b**) In vitro GST pull-down assays showing the interaction between MaCFEM85 and MsWAK16-ED. (**c**) BiFC assays showed that MaCFEM85 and MsWAK16 interacted when co-expressed in *N. benthamiana* leaf cells. Photos were taken at 30 h after infiltration with laser scanning confocal microscopy of yellow fluorescent protein (YFP) (514 nm excitation and 527 nm emission), bright field microscopy, and merged confocal and bright field. Scale bar = 20 µm. YFP^N^ and YFP^C^ represent pUC-SPYNE and pUC-SPYCE vectors, respectively.

**Figure 3 ijms-24-04037-f003:**
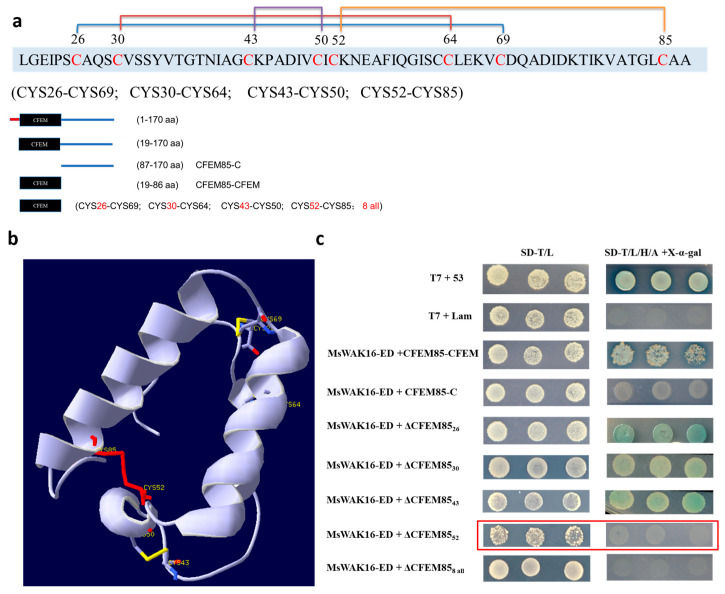
Identification of the interaction site between MaCFEM5 and MsWAK16. (**a**) Schematic illustrations of MaCFEM85. Protein domains in MaCFEM85 were predicted and visualized using SMART (embl-heidelberg.de). (**b**) Positions of the eight cysteines in MaCFEM85 and a schematic showing the four disulfide bonds. (**c**) Truncated MaCFEM85 constructs were generated that included the CFEM domain (19–86 aa of the N-terminal, CFEM85-CFEM) and that included residues 87–170 without the CFEM domain (CFEM85-C). The other five variants were C26A(ΔCFEM85_26_), C30A(ΔCFEM85_30_), C43A(ΔCFEM85_43_), C52A(ΔCFEM85_52_), and C26/30/43/50/52/64/69/85A (ΔCFEM85_8 all_). Each construct was co-transformed in yeast with MsWAK16-ED and grown on either the two-deficiency medium SD-T/L (SD-Trp/Leu) or the four-deficiency medium SD-T/L/H/A + X-α-gal (SD-Trp/Leu/His/Ade + X-α-gal).

**Figure 4 ijms-24-04037-f004:**
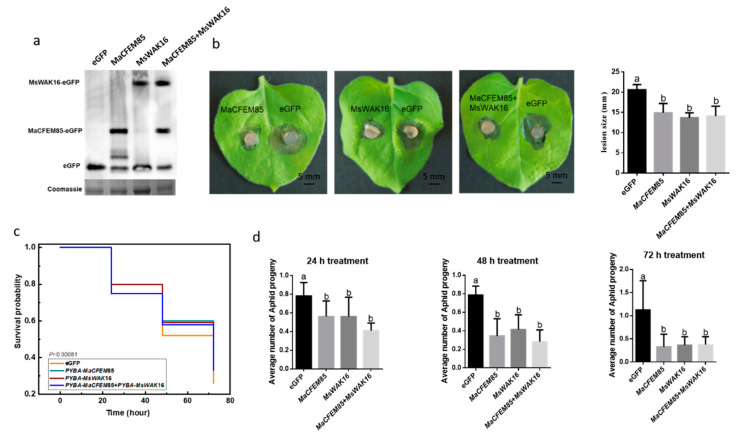
Induction of plant resistance by transient expression of MaCFEM85 and MsWAK16. (**a**) Transient expression of MaCFEM85, MsWAK16, or MaCFEM85+MsWAK16 in *N. benthamiana* leaves. Protein expression level of MaCFEM85 and MsWAK16. Leaf samples were harvested 12 h after infiltration and total soluble protein extracts were prepared. Proteins were separated by SDS–PAGE and analyzed via immunoblot using a GFP-specific antibody. Total protein content (showing equal loading) was measured with Coomassie staining. (**b**) *N. benthamiana* leaf lesions resulting from *B. cinerea* infection. Lesion sizes were measured at 2 d after inoculation in a minimum of nine leaves for each treatment group. Data presented in (**b**) is the mean ± standard error, and Letters indicate statistical significance between groups at *p* < 0.05. (**c**) The mortality risk of aphids feeding on *N. benthamiana* transiently expressing MaCFEM85, MsWAK16, or MaCFEM85+MsWAK16. (**d**) The average number of progeny from each aphid after feeding on tobacco transiently expressing GFP, MaCFEM85, MsWAK16, or MaCFEM85+MsWAK16 for 24, 48, and 72 h, respectively. Values are shown as the mean ± standard error (*n* = 20).

**Figure 5 ijms-24-04037-f005:**
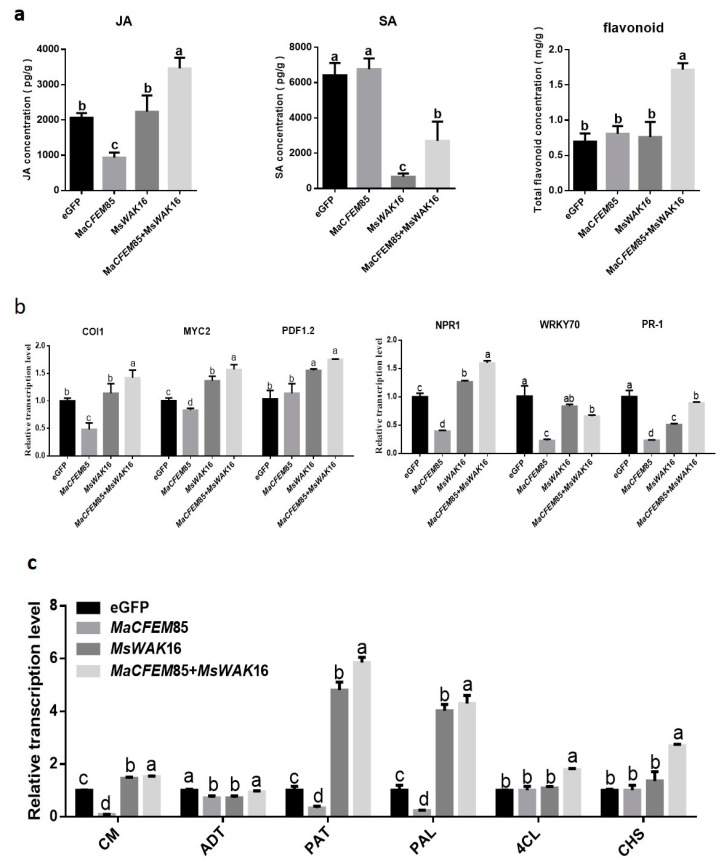
Hormone levels and hormone response gene expression. (**a**) Levels of jasmonic acid (JA), salicylic acid (SA), and total flavonoids in *N. benthamiana* after transient expression of eGFP, MaCFEM85, MsWAK16, or MaCFEM85+MsWAK16 for 12 h. (**b**,**c**) Genes related to plant defense responses were measured in *N. benthamiana* leaves at 12 h after infiltration with Agrobacterium containing plasmids encoding eGFP, MaCFEM85, MsWAK16, or MaCFEM85+MsWAK16. Data are presented as the mean ± standard deviation from three biological replicates. Letters indicate statistical significance between groups at *p* < 0.05. Abbreviations: Chorismate mutase, CM; Prephenate dehydratase, ADT; Aspartate-prephenate aminotransferase, PAT; phenylalanineammonialyase, PAL; 4-coumarate coenzyme A ligase, 4CL; Chalcone synthase, CHS.

## Data Availability

The data presented in this study are available in this article and its Appendix A.

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
