# Peer review of "The Conserved Cysteine-Rich Secretory Protein MaCFEM85 Interacts with MsWAK16 to Activate Plant Defenses"

_ijms, 2023, doi:10.3390/ijms24044037_

Round 1

Reviewer 1 Report

Dear authors, I hope that the following observations are useful.

 • The protein IDs from the databases for MaCFE85 and MsWAK16 are missing . What strain of Metarhzium did you use, do you only mention the species, is its molecular identification in a database?

 • Regarding the cellular localization of MaCFE85: Why was the localization carried out in N. benthamiana? Why was it carried out in leaves? The fungi of the genus Metarhizium present a greater association in the roots of the plants. The characteristics and proteins of the cell walls and membranes of the leaves and roots are different. In figure 1b. colocalization looks similar with fluorescent proteins and when using eGFP-fused or mCherry protein MaCFEM85.

 • Damage analysis using B. cinerea on N. benthamiana leaves: It is not clear if the interaction of the MaCFEM85 and MsWAK16 proteins activate defenses in the N. benthamiana leaf, it may also affect the growth of B. cinerea directly.

 • Why Metarhizium null mutants were not generated in the MaCFE85 gene?.

Reviewer 2 Report

This manuscript details investigation into the fungal genus of Metarhizium Sorokīn. Specifically, the authors look into the entomopathogenic fungus, Metarhizium anisopliaei and identify a novel CFEM protein in the fungus, MaCFEM85 that interacts with a wall kinase of alfalfa plants, MsWAK16 to promote defense responses against fungal infection and insect attack. The authors found that this interaction reinforces plant immune responses, likely due to the triggering of the jasmonic acid signaling pathway. 

This manuscript has a detailed introductory section that situates the reader into the current functions of Metarhizium Sorokīn in plant species. The overall manuscript is well written and easy to follow. The authors’ use of experimental procedures fits nicely for what questions they are asking. However, the authors should include introductory information on MsWAK16 prior to speaking on the elevated expression in the results section in lines 118-119 as it is not clear why MsWAK16 was chosen for downstream experiments. Some questions to consider about this protein is how does MsWAK16 look like? What is the functionality of MsWAK16? The authors also need to include references for MsWAK16. 

Below are comments that should be reviewed before publication. 

The claims about MaCFEM85 localization in line 116 has already been published. Previous findings from reference 31 show this data already. The authors should amend to include that this is previously found information or remove from the manuscript. 

If the authors make the claim that the “interaction of MaCFEM85 with MsWAK16 activates the plant immune response” explain how WAK16 by itself is resistant to B. cinerea. These claims coupled with the experiments conducted in Figure 4 are not clear. The authors should consider that the overexpression of MsWAK16  may lead to misregulation within the plants. From the data, this seems to be an artifact due to the overexpression of MsWAK16.  

As the authors present changes in gene expression of JA and SA genes following overexpression of MaCFEM85 and MsWAK16, the authors should include findings on expressional changes of ethylene-related genes as well, especially considering that “B. cinerea was inhibited by JA and ET accumulation”. 

Why did the authors not use M. anisopliae cultures during infection with B. cinerea to validate the claims that this fungus is acting as an endophyte in plants against fungal pathogen infection. 

Lines 42 and 57: Authors have already introduced SA and JA abbreviations in earlier section, no need to repeat further down in introduction. 

Line 81: Space is needed before reference 26. Review to makes sure this formatting is consistent throughout the text. 

Figure 1A should be moved to the left to not interfere with other figures or remove border of phylogenetic tree. 

Lines 201-202: B. cinerea should be italicized in the title. The same should be done with N. benthamiana in the title at lines 216-217.

Round 2

Reviewer 1 Report

Dear authors, thank you for clarifying the observations.

Reviewer 2 Report

All my comments were addressed by the authors in their responses. I think this revised version has met the requirements of this Journal.